# Optimization of Load Balancing and Task Scheduling in Cloud Computing Environments Using Artificial Neural Networks-Based Binary Particle Swarm Optimization (BPSO)

Mohammed I. Alghamdi 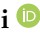

College of Computer Science and Information Technology, Department of Engineering and Computer Science, Al-Baha University, Al-Baha City 1988, Saudi Arabia; mialmushilah@bu.edu.sa

**Abstract:** As more people utilize the cloud, more employment opportunities become available. With constraints such as a limited make-span, a high utilization rate of available resources, minimal execution costs, and a rapid turnaround time for scheduling, this becomes an NP-hard optimization issue. The number of solutions/combinations increases exponentially with the magnitude of the challenge, such as the number of tasks and the number of computing resources, making the task scheduling problem NP-hard. As a result, achieving the optimum scheduling of user tasks is difficult. An intelligent resource allocation system can significantly cut down the costs and waste of resources. For instance, binary particle swarm optimization (BPSO) was created to combat ineffective heuristic approaches. However, the optimal solution will not be produced if these algorithms are not paired with additional heuristic or meta-heuristic algorithms. Due to the high temporal complexity of these algorithms, they are less useful in real-world settings. For the NP problem, the binary variation of PSO is presented for workload scheduling and balancing in cloud computing. Considering the updating and optimization constraints stated in this research, our objective function determines if heterogeneous virtual machines (VMs) Phave the most significant difference in completion time. In conjunction with load balancing, we developed a method for updating the placements of particles. According to the experiment results, the proposed method surpasses existing metaheuristic and heuristic algorithms regarding work scheduling and load balancing. This level of success has been attainable because of the application of Artificial Neural Networks (ANN). ANN has demonstrated promising outcomes in resource distribution. ANN is more accurate and faster than multilayer perceptron networks at predicting targets.

**Keywords:** bioinspired; cloud computing; load balancing; particle swarm optimization (PSO); resource utilization; task scheduling

## 1. Introduction

Using techniques from parallel and distributed computing, cloud computing makes shared computer resources accessible to clients via the Internet. The "pay-as-you-go" business model has nearly democratized cloud computing. Cloud providers, service providers, and end users participate in this phase of software deployment. Cloud service providers offer their customers computational capabilities via virtual computers (VMs). Service providers utilize these virtual machines when it comes to application-level client services. Service providers implement task scheduling algorithms to spread client jobs across virtual machines, reduce response times, ensure a high quality of service (QoS), and maximize resources. Because of this, the job scheduling algorithm is a vital part of any cloud architecture. Cloud computing needs adjustments to the several scheduling techniques utilized in various computer environments. It is possible for a scheduling method optimized for a cluster to perform poorly in the cloud. Before the algorithm can deal with the structure of the cloud environment, the method's parts need to be moved into

the problem space. The greater the variety of virtual machines and size of the workloads being managed, the greater the number of available task configurations. Finding the shortest path across all potential permutations is one of the most challenging problems in computer science. Even though metaheuristic algorithms have already been utilized to assist with cloud scheduling, the authors of this work have devised a new load balancing version of the original PSO approach for cloud scheduling. A load balancing method and a metaheuristic algorithm can be used to make both service providers and customers happy (in terms of resource use and user happiness, respectively) (make-span reduction).

*Our Contribution*

1.　Improve load balancing, so requests are distributed more fairly based on the machine's processing capacity. Improved VM load balancing resulted in much more significant time reductions than previous research.
2.　Examine the complete vector of resources (storage, RAM, and bandwidth) rather than just the CPU when determining whether user requests are suitable with VMs. Consequently, our model is more suited for the cloud.
3.　To meet the needs of service providers and customers, there needs to be a fitness function that cuts down on time while using resources better.
4.　Previous approaches to work schedules would be simplified if a single-goal strategy that considered the interests of both service providers and customers was implemented.
5.　As a result, the PSO and load balancing algorithms can be effectively coupled.

The task scheduling algorithms can lengthen the time needed to complete work while simultaneously reducing the throughput of the overall cloud system. In this regard, the goal of cloud computing is to increase overall performance and make better use of the available computing resources in an environment that contains various types of devices. Several different work scheduling techniques, such as the ant colony optimization algorithm (ACO), the particle swarm optimization algorithm (PSO), and the genetic algorithm (GA), are utilized in a cloud computing environment. In this study, to schedule the activities in a load balanced manner, we have linked the ANN technique with the BPSO strategy to create a hybrid method. Our solution outperforms the traditional BPSO task scheduling algorithm by increasing resource utilization by 22% and decreasing mean time by 33%. For this contribution, we have developed a complete literature analysis and a novel load balancing-enabled job scheduling system, as follows: in Section 2, you will find a discussion connected to the BPSO and ANN, in addition to specifics addressing various work scheduling methods for cloud computing. At the end of Section 2, there is also a comparison of the different ways to schedule tasks in the form of a table. Section 3 elaborates on the issue formulation and the ANN-BPSO approach, starting with definitions, an explanation of the BPSO Technique, problem definition, and a discussion of the prospered framework, which consists of BPSO and ANN. The ANN-BPSO model, System model, Inertia weight approach, and suggested Task scheduler are all covered in this part. The experimental setup in the cloud computing environment for task scheduling is discussed in Section 4, along with the results, experimental configuration setup, dataset information, findings, and discussions. Section 5 presents the research's recommended conclusion with the future perspective of task scheduling and load balancing in cloud computing.

## 2. Literature Survey

The hybrid strategy was utilized by Ahmad M. and his co-workers [1] to balance the load in a cloud environment with diversely accessible resources. This method, known as hybrid GA-PSO, allocates jobs to resources in the most effective manner possible. Utilizing genetic algorithm and particle swarm optimization, its objectives are attained. The authors suggest that using Max is less painful and less expensive while balancing the demand on cloud computing infrastructure.

Workflow scheduling was introduced by Nirmala SJ et al. [2] to ensure that crucial scientific procedures on IaaS clouds are planned. Therefore, workflow scheduling systems

that utilize Catfish's particle swarm optimization (PSO) are more efficient and consume fewer resources than their competitors. When a large number of jobs are conducted concurrently, the execution of an algorithm consumes a significant amount of resources and takes a long time. These solutions do not account for the necessity of load balancing in the cloud work schedule, which continues to be a widespread problem.

Mishra et al. [3] devised the LB approach by modeling its structure after the characteristics of a flock of birds using the BSO-LB algorithm. Virtual machines (VMs) represent food particles in this scenario, while jobs represent birds. The authors received the datasets utilized for their measurements from the cloudlet-based GoCJ. The authors have drastically reduced the reaction time, allowing for an equal division of effort. The FCFS (First Come, First Serve), SJF (Shortest Job First), and RR (Round Robin) approaches are compared along with the proposed technique.

Muhammad Junaid et al. [4] used a support vector machine to classify the input request. Depending on the categorization of the assignment, it was then assigned to a hybrid metaheuristic approach that combined Ant colony optimization with file type formatting. According to them, the hybrid metaheuristic algorithm they developed can be used to keep cloud systems stable. Using criteria such as service level agreement violations, migration times, overhead times, throughput, and quality of service, they compared the efficacy of the suggested strategy.

A mutation that aids in workload distribution. When Awad AI et al. [5] proposed PSO, they aimed to increase load balancing and dependability while decreasing transmission costs, execution, transmission durations, make-spans, and round-trip times. In this method, each virtual machine will execute a proportional number of tasks to its load. If there are multiple significant projects, it is possible that the cloud system will not be able to correctly balance the load, causing the task to take longer than usual to complete. In addition, they did not consider the amount of time and money clients would need to implement their ideas. Arabnejad H et al. [6] found that although these algorithms achieve good outcomes, their great temporal complexity makes them less suitable for real-world computers. In their investigation, Shabnam et al. [7] utilized the bat algorithm. For load balancing and virtual machine optimization, a hybrid strategy is required. This swarm-based approach has been created and implemented to optimize the load on the virtual machine, ultimately resulting in the load on the actual computer being balanced. Yousef Fahim et al. [8] propose a metaheuristic bat technique for assigning virtual machine work.

In Kruekaew B et al. [9]'s implementation of a hybrid approach, an ABC algorithm and a heuristic technique are merged, which can be regarded as a hybrid approach. Considering the time required to develop and distribute the load is essential. It is possible to demonstrate that this method is effective in either homogeneous or heterogeneous contexts. Using this method, we could significantly minimize the number of conflicting factors previously investigated. This algorithm may have been evaluated against a dataset from the real world by assessing several different qualities of service criteria, such as resource consumption, reaction time, etc.

Mala Yadav et al. [10] have developed a multi-criterion scheduling technique for multiprocessor computer systems. This application combines quantum computing and the gravitational search approach, both of which were inspired by nature. This study considers both homogeneous and heterogeneous conditions to determine whether the proposed strategy is beneficial. According to the findings, it produces better-than-anticipated results for the various scheduling objectives, such as load balancing, resource usage, and make-span. This method may have been enhanced in terms of consuming less energy and for workflow applications.

Mala Yadav and colleagues [11] conceived a hybrid metaheuristic algorithm. This system is composed of the genetic and particle swarm optimization algorithms. The primary objective of this study is to find a solution, or at least an approximation of a solution, to the problem of load balancing between virtual machines. The authors assert that the findings obtained from the tests were the best that could have been obtained under

the conditions. Banerjee S et al. [12] designed load balancing to distribute cloudlets (tasks) around virtual machines (VMs) based on each machine's capability, hence decreasing task completion time as well as the makespan of VMs and hosts in the data center. This strategy assigns large workloads (those with a significant size) to VMs that are already available. However, prioritizing large tasks can significantly increase the delay for many minor actions (small jobs), resulting in a significantly longer overall completion time. The cost of execution and resource utilization have not been recognized as indicators of the essential quality of service by cloud providers and clients. Sequentially, many algorithms restrict themselves to a limited set of resources and job sizes to offer an optimal result (large and small tasks). By adopting a BPSO-based task scheduling technique, we provide an initial population and target function more suited to the work at hand and context. Our approach reduces the time required to finish a make-span, while enhancing resource consumption and existing model summary outcomes, as illustrated in Table 1.

**Table 1.** Summary of the related work.

| References | Applied Method | Approach | Advantage | Drawback |
|---|---|---|---|---|
| [13] | The new strategy for task allocation | Provision of load balancing by using task allocation strategy | • Minimizing the VM make-span <br> • Reducing the time of task completion | • Not scalable <br> • Significance degree of the metrics is not used |
| [14] | Improved weighted round robin algorithm | Task based Load Balancing | • Processing time <br> • Effective resource utilization | • Schedule nonredemptive dependent tasks to the VMs <br> • Significance degree of the metrics is not used |
| [15] | SPV-based PSO algorithm | Migration of tasks requiring computing intensity to highperforming computer | • Minimization of processing costs <br> • Minimization of transfer time <br> • Feasibility in heterogeneous systems | • Scalability is not provided <br> • The metrics' significance level is not utilized |
| [16] | PSO | Allocation of extra tasks causing overload to correspond VMs | • Minimal task execution time <br> • Limited task transfer time | • Mutual independence-based tasks execution <br> • Metrics' significance level is not utilized |
| [17] | Genetic algorithm | Allocation of extra tasks causing overload to corresponding VMs | • Task transfer time <br> • Execution costs <br> • Minimum Energy consumption usage <br> • Minimal length of task sequence | • Large time spent on task scheduling and low speed <br> • Significance degree of the metrics is not used |
| [18] | Honeybee algorithm | It models the nutrition behaviour of honeybees | • Minimization of time spent of VM wait time of the task <br> • Expanding the transfer bandwidth <br> • Consideration of sequence priority in VM task sequence <br> • Prolonging the response time <br> • Minimization of make-span | • Migration of mutually independent tasks <br> • Insufficient level of scalability <br> • Significance degree of the metrics is not used |

## 3. Proposed System

The BPSO method, which employs an approach-based binary particle swarm optimization method, can be used to plan and balance a large number of heterogeneous virtual machines rapidly and efficiently [19–29]. First, we describe a new BPSO for task scheduling

and load balancing in cloud computing. Finally, we illustrate and assess the algorithm's temporal complexity.

### 3.1. The Following Definitions Are Included in This Section

What is the meaning of "to start"? It is a graphic device. A tuple can be thought of as the fundamental unit of virtual machines (VM). The ID of a virtual machine is its number of processing elements (PEs), the MIPS is its execution speed per PE, and the prenumber is the number of PEs within the virtual machine. VM is the machine identification number for a virtual machine [30–35].

**Definition 1** (Task). *To determine how many PES are required to accomplish a task on a suitable virtual machine (VM), you must know the task's ID (id), length (in millimeters imperial), and the number of PE (the number of PE needed to run a job on a VM).*

**Definition 2** (optimal solution). *To maximize your resources, your running time, and your overall cost, you must plan in advance. This study found that different sets of tasks should be split up among many different types of virtual machines (VMs).*

**Definition 3** (degree of imbalance" (DI)). *To determine how evenly your virtual machines are used [36–42]. Therefore, a lower DI value demonstrates that the load is more evenly distributed. The formula Equation is utilised to compute the DI (1).*

$$DI = \frac{M_{\max} - M_{\min}}{M_{avg}} \tag{1}$$

Maximum and minimum execution times, together with an overall average, are reported for each VM.

**Definition 4** (the length of time a virtual machine exists). *This is the overall amount of time required for a virtual machine (VM) to complete all of its operations [43–48]. Equation (2) indicates that it is denoted by the letter "CM". n is the number of virtual machines [12] and jobs that can be executed concurrently (2).*

$$CM_i = \sum\nolimits_{j=1}^{n} \frac{M_j.Length}{vm_i.pesnumber \times vm_{i.}mips}, with\ i \in \{1, 2, \ldots m\} \tag{2}$$

**Definition 5** (make-span). *Each task includes a "make-span" that indicates how long it will take to accomplish all of them [5]. Therefore, the scheduler is doing an excellent job of allocating workloads to resources. EQ summarizes it thus, (3):*

$$makespan = max_{1 \le i \le m}\{CM_i\} \tag{3}$$

**Definition 6.** *Utilization of resources is a performance metric that measures the workplace use of resources. If cloud service providers have a high rate of resource usage, they can optimize their earnings. Equation (4) computes the RL [17], or resource consumption:*

$$RL = \frac{\sum_{i=1}^{m} CM_i}{makespan \times i} \tag{4}$$

**Definition 7** (execution expenditure). *Execution costs (EX) are the fees a cloud user pays to the cloud provider for the utilisation of resources to meet a project's objectives. Most consumers must utilise cloud computing in an efficient manner with a short payback period. We can determine the execution cost using Equation (5), and algorithm notations are provided in Abbreviations.*

### 3.2. Using BPSO to Schedule Tasks

In the context of BPSO work scheduling, particles are characterized as matrices, and one feasible solution is task distribution among heterogeneous virtual machines using a m × n position allocation matrix. In the matrix, columns represent job allocation and rows represent jobs assigned to virtual machines. A single virtual machine is required for each task, and the number 1 in each column indicates that a particular task has been given to a virtual machine, whilst zero indicates that no work has been assigned. Abbreviations demonstrates that there are three virtual machines, each with seven workloads. Similar to the location matrix, the velocity matrix for each particle has an element range of [0, 1]. To illustrate the ideal method for distributing jobs across heterogeneous virtual machines, we can use the notations "pBest" and "gBest", which are matrices of zero and one, respectively.

### 3.3. Binary Particle Swarm Optimization

Using metaheuristics, optimization problems involving discrete functions such as integer programming, scheduling, and routing can be resolved. Eberhart and Kennedy conceived and suggested binary particle swarm optimization (BPSO), which examines the binary search space to determine what exists. In an effort to create a middle ground between exploration and extraction, BPSO blends local and global search algorithms. All particles' velocities are continuously updated by combining the particle's position with the best possible location for that particle (pBest), and the greatest possible location globally (pGlobal) (gBest).

Unlike PSO, which changes the position of each particle based on its current position and velocity, BPSO uses only the particle's current velocity. The sigmoid function is widely applied for updating particle locations. Binary numbers can also be used to demonstrate where and how fast particles in a population are moving.

### 3.4. Problem Description

Consider swarm optimization with p particles, m VMs of heterogeneous types, and n distinct forms of workload (T). This matrix, expressed as (m × n) = (m + n), depicts the distribution of jobs across these virtual machines. This is a position allocation matrix for a particle k that specifies where the task corresponds to (5).

$$PA^K = \begin{array}{c} Vm_1 \\ Vm_2 \\ . \\ . \\ . \\ Vm_m \end{array} \left[ \begin{array}{c} x_{1,1}x_{1,2}x_{1,3}\ldots\ldots\ldots x_{1,n-1}x_{1,n} \\ x_{2,1}x_{2,2}x_{2,3}\ldots\ldots\ldots x_{2,n-1}x_{2,n} \\ \ldots\ldots\ldots \\ \ldots\ldots\ldots \\ \ldots\ldots\ldots \\ x_{m,1}x_{m,2}x_{m,3}\ldots\ldots\ldots x_{m,n} \end{array} \right] \tag{5}$$

$$where \; x_{ij} = \quad \{1 \; if \; T_j \; is \; assigned \; to \; Vm_i$$
$$0 \; else$$

Each particle in the swarm represents a possible solution that could lead to a viable solution. In other words, this indicates that the perfect solution may exist within the subatomic particle k. Therefore, in this procedure, all particles are subjected to the same number of iterations prior to the identification of a particle that provides the expected ideal response, based on repeated comparisons between these particles. As illustrated by the equation, one technique to expedite the procedure is to iterate each particle the same number of times (5). As a meta-heuristic algorithm, BPSO becomes less applicable in real-world circumstances. If you lack a particle that can produce an optimum response, you cannot guarantee an ideal outcome. The distinct activities and three virtual machines tasks are provided in Table 2. This work therefore contains two sub-problems, which are as follows [40–48]:

1.  How to use the BPSO method to organize and balance different kinds of jobs on different kinds of virtual machines in the cloud.
2.  How can the time complexity of BPSO be reduced so that it can be used in real-world situations?

**Table 2.** A particle k for seven distinct activities and three virtual machines.

|      | Task 1 | Task 2 | Task 3 | Task 4 | Task 5 | Task 6 | Task 7 |
|------|--------|--------|--------|--------|--------|--------|--------|
| VM1  | 0      | 1      | 0      | 0      | 1      | 0      | 0      |
| VM2  | 1      | 0      | 1      | 0      | 0      | 0      | 1      |
| VM3  | 0      | 0      | 0      | 1      | 0      | 1      | 0      |

### 3.5. Scheduling and Load Balancing via Binary Particle Swarm Optimization

Here, we elaborate on a new, low-complexity, low-cost BPSO for scheduling and balancing heterogeneous cloud-based virtual machine operations. First, we explain the suggested BPSO framework and define the problem-related objective function in order to be more specific. To accelerate research into binary space, we have established a revolutionary concept and calculation for each particle in our model. This provides for substantial time savings while delivering the highest quality solution available. In light of this, we present two constraints, namely update and optimization constraints, to determine how many fitness solutions can quickly generate an ideal solution [49–53]. The final step in our effort to reduce simulation costs is to develop a more accurate model of particle position, based on a load balancing method with an updating constraint. Our fitness evaluation technique has been improved to prevent particles from wandering too far from the optimal response.

### 3.5.1. The BPSO Framework

BPSO can be utilised to address the problem of job scheduling and load balancing utilising the cloud computing paradigm depicted in Figures 1 and 2. These are the three components that comprise the cloud system; the demands of users are divided into several tasks and sent to the cloud management as the initial module. For each virtual machine, cloud management (CM) generates a local work queue (VM). IBPSO-LBS, together with our pricing model and mapping, is a submodule of CM. IBPSO-LBS schedules all jobs across heterogeneous VMs based on updating and optimization constraints [54–59]. Mapping assigns each local queue to a VM, and the price model determines the execution cost for all user tasks. A virtual machine manager for many hosts concludes the third module. Numerous virtual machines are deployed for various purposes.

### 3.5.2. Objective Function

Virtual machines (VMs) running on a variety of hosts and performing a range of tasks each have their own unique completion time, according to Equation $CT_i$ (2). The completion time difference of m diverse projects is defined as:

$$dct = \left| CT_i - CT_j \right| \tag{6}$$

where $i$ and $j$ have values between 1 and m, and $i$ and $j$ are not equivalent. dct is meant to identify virtual machines that are overcrowded or underloaded. As indicated in Equation (6), the objective function attempts to reduce the disparity between available heterogeneous VMs and overall completion time, while reducing user task waiting times. This strategy simultaneously reduces time and metrics.

As a result, Equation (7) represents the objective function:

$$f(CT) = \max(CT_i - CT_j \big| / 1 \le i \big| j \le m \}$$

where CT:

$$(CT_1, CT_2, CT_3, \ldots CT_m)dct_{max} \tag{7}$$

As restrictions for updating and optimising, the sigmoid function and the minimum completion time for unique VMs are utilised, respectively. The computation of these constraints is detailed in Section 3.5.4.

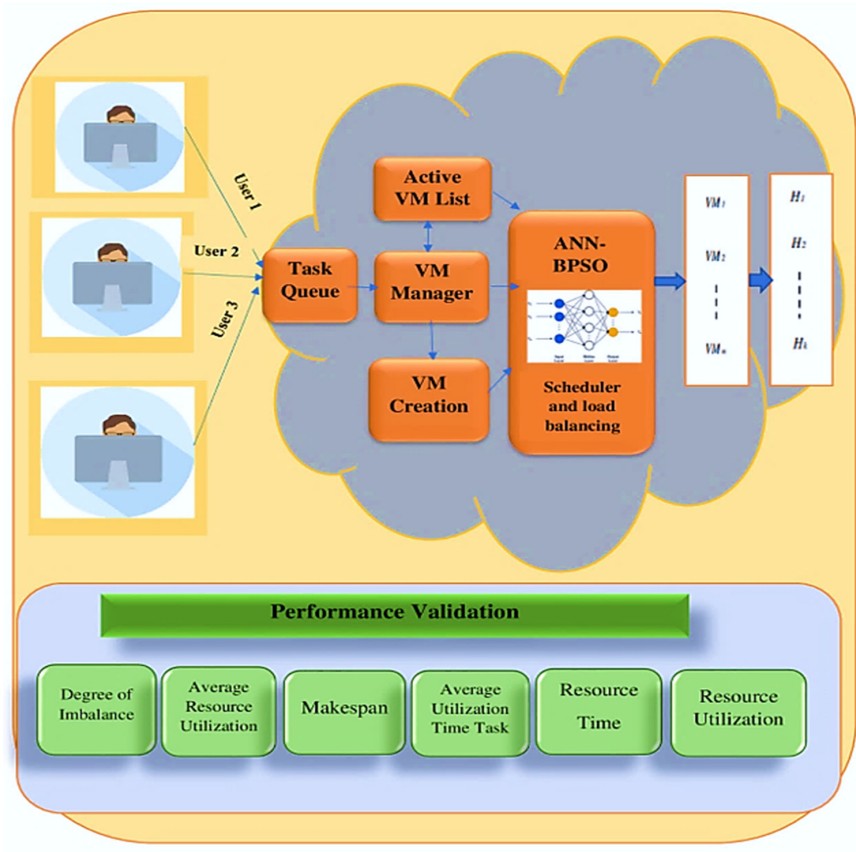

**Figure 1.** Proposed ANN-BPSO system architecture.

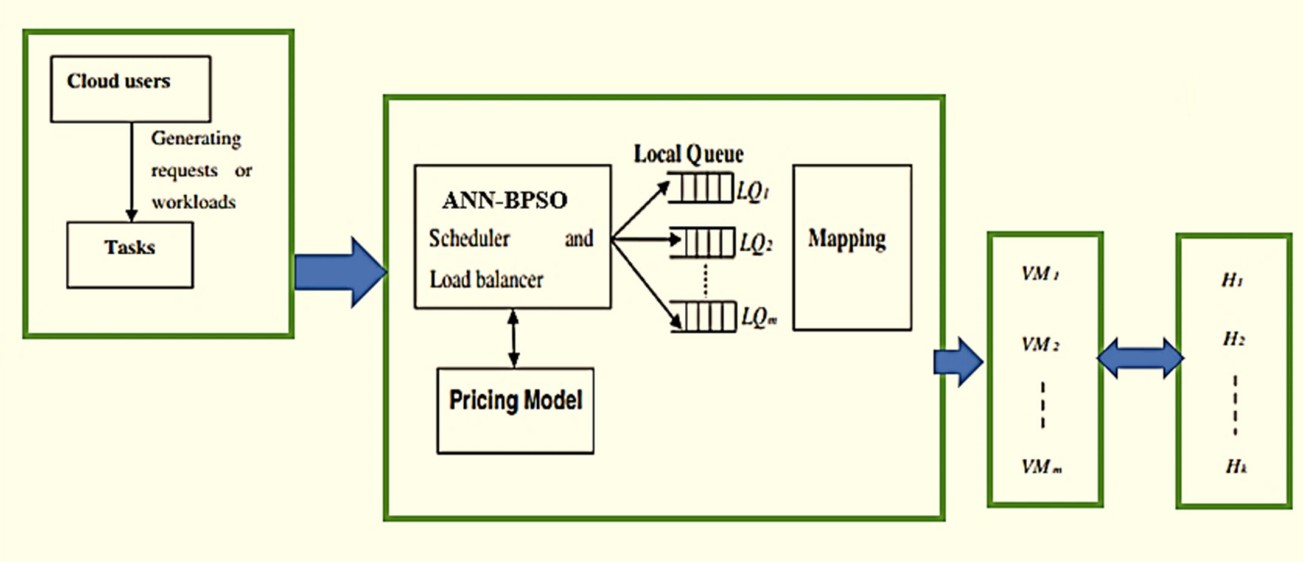

**Figure 2.** BPSO Framework.

### 3.5.3. Definition of a Particle in Context

This stage of the study aims to identify a reference particle in order to expedite the search in binary space and permit convergence on an ideal solution. This value has been calculated using the output of the sigmoid function. As a starting point, the researchers determine the potential solution of a particle.

S is utilized to transform values between zero and one in the velocity $(V_k(i, j))$ of each k-particle, as demonstrated in Equation (8). Using the transformed data, it is feasible to anticipate which virtual machine will be assigned to which task. Negative velocity values are applied first, followed by the multiplication of random values in Equation (8), such that the usable value of 1 is preserved:

$$S(V_k(i,j)) = \frac{1}{1+\exp(-V_k(i,j)*r)} \, with \, i \in \{1, 2, \ldots, m\} \\ and \, j \in \{1, 2, \ldots, n\} \tag{8}$$

The $i$th row and jth column element of the kth velocity matrix ($n$) illustrate the number of different kinds of virtual machines ($m$) and jobs ($n$). In Equation (9), all of the up-to-date values from each VM are utilised to determine the normalization coefficient for a particle k:

$$z_i{}^k = \sum_{j=1}^{n} S(V_k(i,j)) \tag{9}$$

The normalisation coefficient of particle $k$ in the $i$th row is. It is possible to determine the particle $k$'s maximum and minimum normalising coefficients from Equation (9) and its average and intermediate normalising coefficients from Equations (10)–(13). It is essential to differentiate between average and intermediate normalising coefficients, as they correspond to distinct value ranges between the two extremes:

$$sig^k{}_{max} = \max{}^k{}_{1 \leq i \leq m}(z^k{}_i) \tag{10}$$

$$sig^k{}_{min} = \min{}^k{}_{1 \leq i \leq m}(z^k{}_i) \tag{11}$$

$$sig^k{}_{avg} = \frac{sig^k{}_{max} + sig^k{}_{min}}{2} \tag{12}$$

$$sig^k{}_{int} = \frac{sig^k{}_{max} + sig^k{}_{avg}}{2} \tag{13}$$

There are a few ways to calculate the updating coefficients for Equations (9)–(12). The most important is to use Equations (14)–(16):

$$vc^k \frac{sig^k{}_{max}}{l}{}_{max} \tag{14}$$

$$vc^k{}_{avg} = \frac{sig^k{}_{avg}}{l} \tag{15}$$

$$vc^k \frac{sig^k{}_{int}}{l}{}_{int} \tag{16}$$

Calculating optimization coefficient and update coefficient based on Equations (14)–(16), we may determine the interval [0, 1] for these coefficients. Equations (17) and (18) describe how to execute two subtractions in order to arrive at the best coefficients:

$$xc = vc^k{}_{max} - vc^k{}_{avg} \tag{17}$$

$$yc = vc^k{}_{max} - vc^k{}_{int} \tag{18}$$

$$vc^k = \frac{xc + yc}{2} \tag{19}$$

$$qc^k = xc + y \tag{20}$$

A particle's influence on a future solution may be measured using the updating and optimization coefficients in Equations (19) and (20).

### 3.5.4. Execution Time for the Gap

**Definition 8.** *In this study, the phrase "gap execution time" refers to the maximum execution time difference ($dct_{max}$) that can lead to an optimal solution, as determined by the capacity of a particle. The optimization and update coefficients determine this capacity. Constraints based on optimization and updating coefficients are employed to establish two optimization and updating constraints, which correspond to the longest and shortest gap execution periods, as illustrated in Equations (21) and (22):*

$$get^k k_{minmax} \tag{21}$$

$$get^k k_{minmin} \tag{22}$$

With the help of Equations (19) and (20), we can determine which heterogeneous VM has the fastest completion time for particle k and a certain update.

Particle position updates are controlled by defining a minimum gap execution time in Section 3.5.6. Because particles could get stuck in local optima and move too far away from the best solution, the maximum gap execution time has been set. Based on our calculations of the optimization and updating coefficients, we can identify two possible ranges of optimal fitness solutions. In the first scenario, it is impossible to discover a fitness solution that is smaller than or equal to the minimum gap execution time. Local optima are difficult to achieve because particles can become trapped in them. Consequently, the total running duration is greatly increased.

The second constraint on the fitness solution is the minimum and maximum gap execution lengths. Local optimum conditions are insufficient to hold these particles. With less time and a smaller window, scheduling efficiency increases.

### 3.5.5. Particle Velocity Is Updated

At each iteration *t*, the velocity matrix of a particle is updated using the following Equation (23):

$$\begin{aligned} V^{t+1}{}_k(i,j) &= w \times V^t{}_k(i,j) + d_1 \times e_1 \\ &\times \left( pBest^t{}_k(i,j) - X^t{}_k(i,j) \right) + d_2 + e_2 \\ &\times \left( pBest^t{}_k(i,j) - X^t{}_k(i,j) \right) \end{aligned} \tag{23}$$

where $w$ is the inertia factor influencing the local and global abilities of the algorithm; $\forall(i.j), i \notin \{1, 2, \ldots, m\}$ and $j \in \{1, 2, \ldots, n\}$, $V^t{}_k(i,j) \notin [0, 1]$ and $X^t{}_k(i,j) \notin \{0,1\}$ are the element in ith row and jth column of the kth velocity matrix and the element in ith row and jth column of the kth position matrix at iteration t, respectively; a random number is defined in terms of its position in space (0 or 1), whereas the weights impacting cognitive factors (D1) and social factors (D2) are defined in terms of their values (0 or 1). $pBest^t{}_k$ and $gBest^t{}_k$ represent particle k's global and local best position, respectively.

### 3.5.6. Interia Value

LDDIW is used as the starting point for iterative inertia weighting, which is implemented as a variable parameter here. An example of this may be found in Equation (24), below:

$$y = y_{\max} + \frac{(y_{\min} - y_{\max})}{iter_{\max}} \times iter \tag{24}$$

For the BPSO algorithm, the maximum and minimum inertia weights can be found here $y_{max}$ and $y_{min}$ here. iter and $iter_{max}$ denote the current and maximum iteration time, both of which are set to 100. The best values are between 0.9 and 0.5.

### 3.6. Proposed ANN-BPSO Algorithm

The suggested ANN-IBPSO algorithm, as depicted in Algorithm 1 and Figure 3 is described in this section.

---

**Algorithm 1.** Pseudocode of ANN-BPSO

---

1. Initiate the position and velocity vectors of each particle.
2. Create a discrete version of the continuous position vector.
3. Calculate the fitness value of each particle using a fitness function.
4. This is the first time that the best location has been allocated to a particle's "psobest". Instead of using "pbest", use the particle's current position value instead if its current fitness value is higher than its "psobest".
5. The particle with the highest fitness value should be chosen as the best.
6. Change the vectors of each particle using the following equations:

$$Vj + 1 = qVj + i1n1 * (psobest - xj) + i2n2 * (psobest - xj), jits foriteration \quad (25)$$

$$Xj + 1 = Xj + Vj + 1 \quad (26)$$

Since the acceleration coefficients are n1 and n2, we can use them to produce random values for psobest, which is the optimal position for all of the particles in our population.

7. Once the termination condition has been met, return to step 2 and repeat step 7. Once all iterations have been completed, we have reached the terminating condition when there is no further change in particle fitness.
8. Finally, the best particle is produced.

---

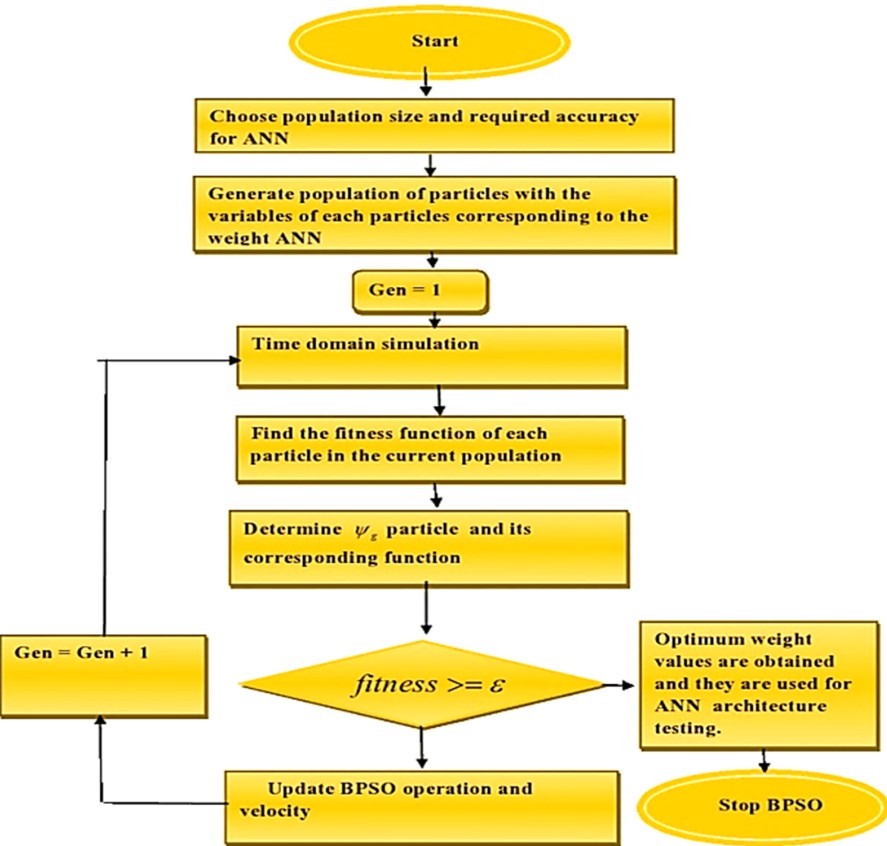

**Figure 3.** Flowchart for ANN using BPSO algorithm.

Using a binary PSO algorithm, Jean Pepe Buanga Mapetu and his coworkers developed their method. We require an immediate solution for the scheduling and balancing of a large number of diverse VMs. Utilizing an appropriate BPSO algorithm architecture, task scheduling and load balancing are performed. In this structure, there are three components. In the first module, user requirements are broken into individual jobs. This responsibility falls to the cloud administrator. The Cloud Manager, the second module, creates a local queue for each virtual machine. In this module, the BPSO approach is used to balance work by assigning each local queue to the appropriate virtual machine, taking optimization constraints into account. In addition, the pricing model takes the whole cost of the completed work into account. Three modules rely on many hosts; the virtual machine manager keeps track of all the virtual machines that are being utilized to accomplish user-initiated tasks. This method is scalable if executed correctly.

### 3.7. Analysis of Complexity

The ANN-BPSO method proposed in this study may be divided into two fundamental phases: the initialization of particles and velocity, and updates to particle positions and velocities, as well as fitness solution evaluations. Before estimating the algorithm's time complexity, it is crucial to determine how long it takes to complete each stage. Position and velocity must be initialized for each particle in a swarm optimization of p particles. Each particle can be viewed as having a $n * n$ matrix of virtual machines (VMs). This calculation involves O particles, O virtual machines, and O tasks (pnm).

Additional rounds of t (the number of iterations) are employed to analyse the position and velocity of each particle in order to find a fitness solution. This phase's time complexity is determined by the number of iterations, particles, tasks, and virtual machines (VMs). In this circumstance, you must fight with these two extremes. Consider the number of repetitions. If we wish to identify the ideal solution, we must determine how many iterations and particles we skipped during our search. Temporal complexity is equal to zero because all iterations and particles have concluded (spnm). In an ideal situation, x, s, and y should be satisfied quickly by line 2 and 5 of Algorithm 1 O (s-x) (p-y) nm is the new temporal complexity's new complexity.

## 4. Result and Evaluation

In this section, we compare our proposed ANN-BPSO method to the state-of-the-art scheduling algorithms already in use, and we explain the experimental findings and evaluation of its performance. Five other scheduling algorithms were compared to our suggested algorithm to see how well it worked in terms of make-span, average waiting time, response time, and resources used.

### Environmental Setup

It is difficult to try out new approaches or ideas when the infrastructure is inflexible, as it often is in a cloud computing environment like Amazon EC2 or Microsoft Azure, because of issues like security, speed, and the high cost in currency of repeating testing [60–64]. These sorts of tests are difficult to execute on real-world cloud infrastructures since they need a lot of effort to make them scalable and repeatable.

According to previous research, the CloudSim-3.0.3 simulator may be used to assess a proposed algorithm's performance in real-world scenarios. Additional data was created at random by the simulator itself and used in the testing. It was possible to write all of the Java code and execute it concurrently on an Intel Core i5-6500 PC running at 3.2 GHz with 4GB of RAM. Table 3 lays out the components and parameters of our model. First, the experiment on independent jobs and the impact of update coefficient, and second, experiments on varied workloads and number of virtual machines are related to these factors in CloudSim parameters [65,66]. Since all VMs are spread evenly across all hosts, the first value represents how things were originally set up. The VMs are distributed unevenly between hosts, which is reflected in the second result.

**Table 3.** Degree of Imbalance (DI) analysis of ANN-BPSO technique with existing methods.

| | Degree of Imbalance (s) | | | | | |
|---|---|---|---|---|---|---|
| N. of Tasks | Heuristic | Meta-Heuristic | PSO | IBPSO-LBS | Heuristic-FSA | ANN-BPSO |
| 1000 | 3.678 | 2.275 | 1.985 | 0.957 | 0.197 | 0.0365 |
| 2000 | 3.875 | 2.945 | 2.256 | 1.234 | 0.214 | 0.0628 |
| 3000 | 3.987 | 2.845 | 2.578 | 1.856 | 0.296 | 0.8751 |
| 4000 | 4.238 | 3.214 | 2.865 | 1.334 | 0.398 | 0.0911 |
| 5000 | 4.967 | 3.546 | 2.983 | 0.987 | 0.324 | 0.0998 |

As a way to test our proposed ANN-BPSO method, we established a task size of 100,000 to 600,000 MI for a single job, and between 1000 and 5000 tasks can be assigned at a time, with the number of heterogeneous virtual machines being constant at 150. Each of the 15 hosts had an equal number of virtual machines. Under capacity limits, more tasks means more incoming user tasks, which is what we are testing in this study. This experiment employs a large number of tasks to find out how well ANN-BPSO performs in a scenario where the number of tasks increases but all hosts have about the same number of VMs and overall processing speeds. Below, Table 3 illustrates the outcomes of our algorithm in contrast to other current algorithms such as Heuristic, Meta-Heuristic, PSO, and IBPSO-LBS, in terms of degree of imbalance, average result utilization, average waiting time tasks and resource utilization, and make-span and reaction times.

As demonstrated in Figure 4, the ANN-BPSO method has a lower degree of imbalance than the Heuristic-FSA, Heuristic, Meta-heuristic, PSO, and IBPSO-LBS methods. As a result, the proposed approach outperforms the other current methods in terms of load balancing. For 1000 tasks considered, the degree of imbalance is 0.0365 s for the ANN-BPSO method, whereas for Heuristic, Meta-Heuristic, PSO, IBPSO-LBS, and Heuristic-FSA the DI is 3.678 s, 2.275 s, 1.985 s, 0.957 s, and 0.197 s, respectively. For 5000 tasks, the DI for the ANN-BPSO method is 0.0998 s while it is 4.967 s, 3.546 s, 2.983 s, 0.987 s, and 0.324 s for Heuristic, Meta-Heuristic, PSO, IBPSO-LBS, and Heuristic-FSA, respectively.

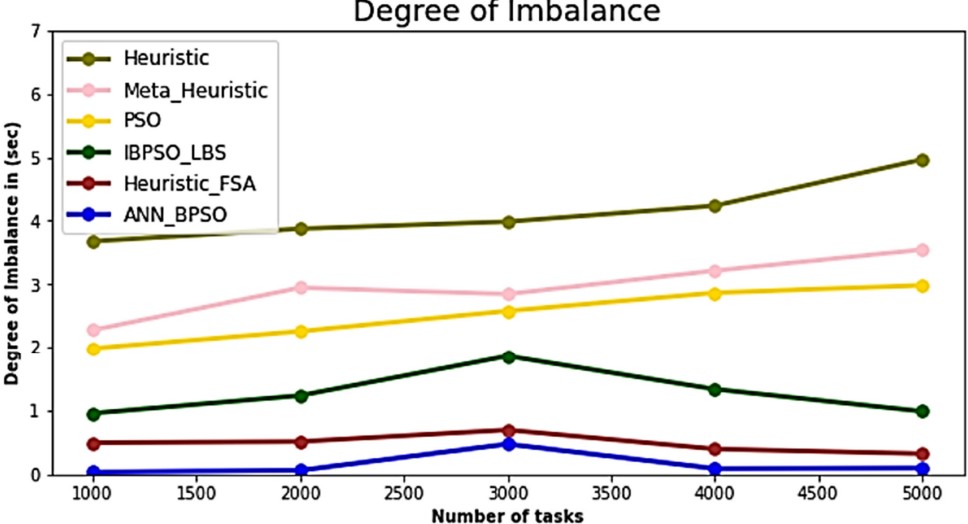

**Figure 4.** Degree of Imbalance (DI) analysis of ANN-BPSO technique with existing methods.

From Figure 5 & Table 4, we can see that, there is a noticeable difference between ANN-BPSO and the other algorithms in terms of the average resource utilisation (RU). Never underestimate the importance of RU. This necessitates a very minimal amount of resource waste. Because of this, the ANN-BPSO method makes optimal use of the various heterogeneous resources readily available. For 1000 tasks considered, the RU is 96.84% for the ANN-BPSO method, whereas for Heuristic, Meta-Heuristic, PSO, IBPSO-LBS, and Heuristic-FSA, the DI is 91.23%, 92.24%, 93.36%, 94.05%, and 95.34%, respectively. For 5000 tasks, the RU for the ANN-BPSO method is 93.56% while it is 87.26%, 88.24%, 89.12%,

90.27%, and 91.91% for Heuristic, Meta-Heuristic, PSO, IBPSO-LBS, and Heuristic-FSA, respectively.

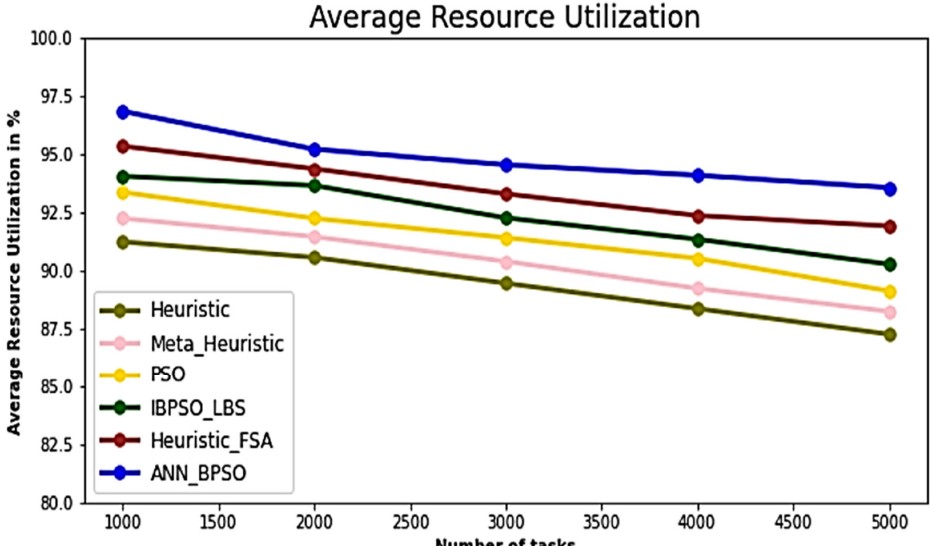

**Figure 5.** Average resource utilization analysis of ANN-BPSO technique with existing methods.

**Table 4.** Average resource utilization analysis of ANN-BPSO technique with existing methods.

| | Average Resource Utilization (%) | | | | | |
|---|---|---|---|---|---|---|
| N. of Tasks | Heuristic | Meta-Heuristic | PSO | IBPSO-LBS | Heuristic-FSA | ANN-BPSO |
| 1000 | 91.23 | 92.24 | 93.36 | 94.05 | 95.34 | 96.84 |
| 2000 | 90.56 | 91.45 | 92.24 | 93.65 | 94.37 | 95.21 |
| 3000 | 89.45 | 90.39 | 91.41 | 92.26 | 93.29 | 94.54 |
| 4000 | 88.36 | 89.23 | 90.52 | 91.34 | 92.36 | 94.09 |
| 5000 | 87.26 | 88.24 | 89.12 | 90.27 | 91.91 | 93.56 |

The make-span of the ANN-BPSO approach in comparison to the other existing techniques is provided in Figure 6 and Table 5. There is a reduction in overall execution time while using our suggested approach for a different job. The make-span for the ANN-BPSO method is 90 units for 1000 tasks considered, whereas the make-span for Heuristic, Meta-Heuristic, PSO, IBPSO-LBS, and Heuristic-FSA is 365 s, 260 s, 150 s, 96 s, and 95 s, respectively. For 5000 tasks, the make-span for the ANN-BPSO method is 150 s, while it is 395 s, 297 s, 200 s, 170 s, and 170 s for Heuristic, Meta-Heuristic, PSO, IBPSO-LBS, and Heuristic-FSA, respectively.

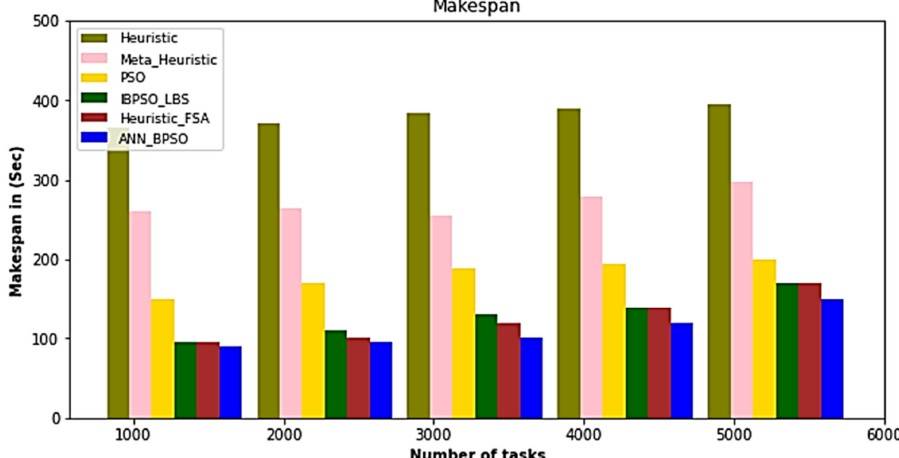

**Figure 6.** Make-span analysis of ANN-BPSO technique with existing methods.

**Table 5.** Make-span analysis of ANN-BPSO technique with existing methods.

| | | | Make-Span (s) | | | |
|---|---|---|---|---|---|---|
| **N. of Tasks** | **Heuristic** | **Meta-Heuristic** | **PSO** | **IBPSO-LBS** | **Heuristic-FSA** | **ANN-BPSO** |
| 1000 | 365 | 260 | 150 | 96 | 95 | 90 |
| 2000 | 371 | 265 | 170 | 110 | 100 | 96 |
| 3000 | 385 | 255 | 190 | 130 | 120 | 100 |
| 4000 | 390 | 280 | 195 | 140 | 140 | 120 |
| 5000 | 395 | 297 | 200 | 170 | 170 | 150 |

In Table 6 and Figure 7, the AWT of ANN-BPSO's is lower than the six other methods. There will be less waiting time for each job to be assigned to a virtual machine, which will result in faster processing times. For AWT, the ANN-BPSO algorithm performs better. For 1000 tasks, the ANN-BPSO method has an AWT of 74 s, whereas the Heuristic, Meta-Heuristic, PSO, IBPSO-LBS, and Heuristic-FSA methods have AWTs of 142 s, 140 s, 135 s, 120 s, and 108 s, respectively. For 5000 number of tasks, the AWT for the ANN-BPSO method is 110 s while it is 159 s, 155 s, 150 s, 161 s, and 134 s for Heuristic, Meta-Heuristic, PSO, IBPSO-LBS, and Heuristic-FSA, respectively.

**Table 6.** Average waiting time task (AWT) analysis of ANN-BPSO technique with existing methods.

| | | | Average Waiting Time Task(s) | | | |
|---|---|---|---|---|---|---|
| **N. of Tasks** | **Heuristic** | **Meta-Heuristic** | **PSO** | **IBPSO-LBS** | **Heuristic-FSA** | **ANN-BPSO** |
| 1000 | 142 | 140 | 135 | 120 | 108 | 74 |
| 2000 | 149 | 145 | 139 | 134 | 115 | 70 |
| 3000 | 150 | 149 | 141 | 146 | 120 | 85 |
| 4000 | 155 | 150 | 145 | 159 | 125 | 90 |
| 5000 | 159 | 155 | 150 | 161 | 134 | 110 |

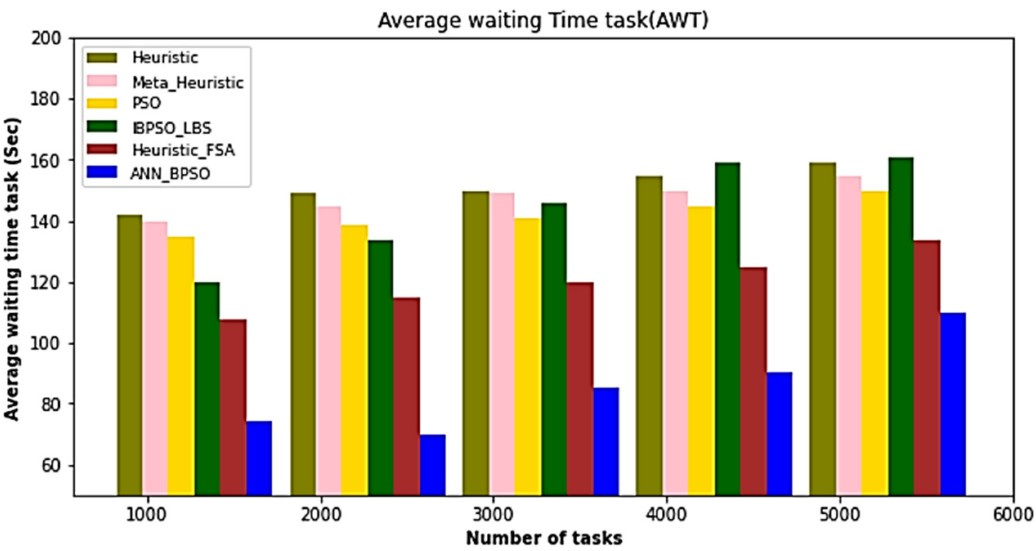

**Figure 7.** (AWT) analysis of ANN-BPSO technique with existing methods.

In Figure 8 and Table 7, the reaction times for various numbers of jobs for the Heuristic, Meta-Heuristic, PSO, IBPSO-LBS, and Heuristic-LBS methods are provided, as well as the proposed ANN-BPSO approach. For the large number of tasks considered, the ANN-BPSO method has a response time of 1.84 s, whereas the Heuristic, Meta-Heuristic, PSO, IBPSO-LBS, and Heuristic-FSA methods have response times of 3.99 s, 3.54 s, 2.89 s, 2.24 s, and 1.99 s, respectively. For 100 tasks, the response time for the ANN-BPSO method is 5.81 s, while it is 9.75 s, 8.73 s, 8.49 s, 7.63 s, and 6.55 s for Heuristic, Meta-Heuristic, PSO, IBPSO-LBS, and Heuristic-FSA, respectively.

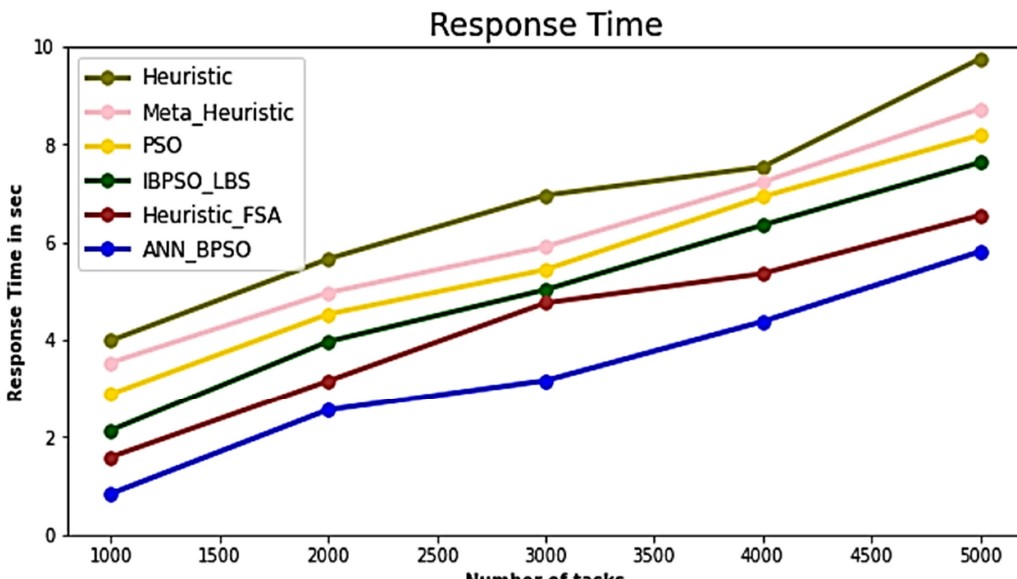

**Figure 8.** Response time analysis of ANN-BPSO technique with existing methods.

**Table 7.** Response time analysis of ANN-BPSO technique with existing methods.

| | | | | | | |
|---|---|---|---|---|---|---|
| **Response Time in Seconds** | | | | | | |
| **N. of Tasks** | **Heuristic** | **Meta-Heuristic** | **PSO** | **IBPSO-LBS** | **Heuristic-FSA** | **ANN-BPSO** |
| 20 | 3.99 | 3.54 | 2.89 | 2.24 | 1.99 | 1.84 |
| 40 | 5.66 | 4.87 | 4.53 | 3.87 | 3.66 | 2.56 |
| 60 | 6.96 | 5.91 | 5.74 | 5.33 | 4.76 | 3.17 |
| 80 | 7.54 | 7.23 | 6.93 | 6.75 | 5.36 | 4.38 |
| 100 | 9.75 | 8.73 | 8.49 | 7.63 | 6.55 | 5.81 |

In Figure 9 and Table 8, the resource utilization of the proposed ANN-BPSO approach with existing methods for various numbers of jobs is illustrated. The resource utilization for the ANN-BPSO method is 0.25 s for the number of tasks considered, whereas the resource utilization for the Heuristic, Meta-Heuristic, PSO, IBPSO-LBS, and Heuristic-FSA methods is 0.72 s, 0.65 s, 0.53 s, 0.43 s, and 0.36 s, respectively. For 50 tasks, the resource utilisation for the ANN-BPSO method is 0.66 s, while it is 0.97 s, 0.93 s, 0.88 s, 0.75 s, and 0.69 s for Heuristic, Meta-Heuristic, PSO, IBPSO-LBS, and Heuristic-FSA, respectively.

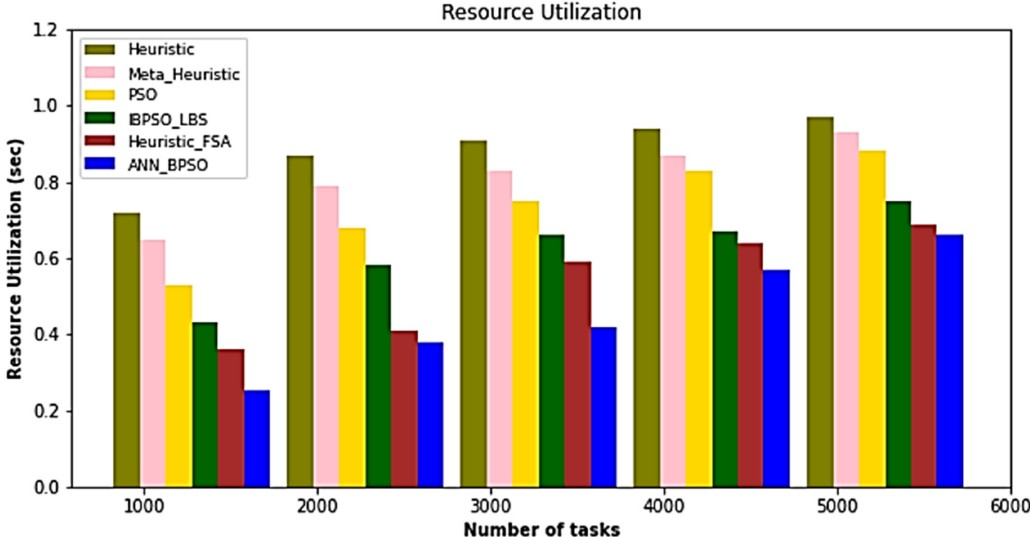

**Figure 9.** Resource utilization analysis of ANN-BPSO technique with existing methods.

**Table 8.** Resource utilization analysis of ANN-BPSO technique with existing methods.

| Resource Utilization in Seconds | | | | | | |
|---|---|---|---|---|---|---|
| N. of Tasks | Heuristic | Meta-Heuristic | PSO | IBPSO-LBS | Heuristic-FSA | ANN-BPSO |
| 10 | 0.72 | 0.65 | 0.53 | 0.43 | 0.36 | 0.25 |
| 20 | 0.87 | 0.79 | 0.68 | 0.58 | 0.41 | 0.38 |
| 30 | 0.91 | 0.83 | 0.75 | 0.66 | 0.59 | 0.42 |
| 40 | 0.94 | 0.87 | 0.83 | 0.67 | 0.64 | 0.57 |
| 50 | 0.97 | 0.93 | 0.88 | 0.75 | 0.69 | 0.66 |

## 5. Conclusions and Future Scope

In order to reduce the amount of time spent waiting for and balancing cloud computing resources, this study created ANN-BPSO, a low-cost binary variant of the well-known PSO algorithm (BPSO). Our solution outperforms the traditional BPSO task scheduling algorithm by increasing resource utilization by 22% and decreasing mean time by 33%. The following adjustments are essential:

- A low-complexity and low-cost load balancing approach based on BPSO is being developed.
- A reference for each particle is being sought to accelerate the search for an optimal solution and the search exploration in binary space.
- The method of updating particle positions about the load balancing strategy is being enhanced to prevent overloaded and underloaded VMs.

BPSO uses two "reference" coefficients: optimization and updating; this is essential to remember. The optimization and updating constraints depend on the maximum and minimum execution time gaps, per this reference. In this system, there are two constraints: an optimization constraint that inhibits the attainment of local optimums and an updating constraint that governs particle location updates about the method and goal of load balancing. By imposing an optimization constraint, it is possible to lower both the running time of the ANN-BPSO method and the waiting time for user tasks. At the same time, an objective function predicts the most significant difference in completion time across all assigned VMs. It has been demonstrated that the ANN-BPSO algorithm outperforms FSA and meta-heuristic approaches such as IBPSO-LB when simulating environmental changes. VMs, distinct tasks (ICTs), and varied workload sizes are examples of ILTs. This unique technique is faster than heuristic algorithms in real-world computing environments with low temporal complexity, which benefits consumers by reducing request wait times. Our proposed method successfully balances the load, schedules work, and makes the system scalable. In the last few years, energy-aware strategies have drawn considerable attention from the research community. However, in our proposed algorithm, we do not have to consider energy consumption, which is one of the critical parameters nowadays. So, in the future, we aim to satisfy both cloud service providers and their customers by including energy usage and live migration factors that significantly impact cloud performance and load balancing. The ANN-BPSO model method will be tested with actual workflows, workloads, and cloud infrastructures. The study of multi-goal objectives will form the basis of our future work. We also aimed to learn more about the dynamic load balancing method to combat this waste of cloud resources. These findings suggest that we should also implement measures to reduce SLA violations to boost service quality.

**Funding:** This research received no external funding.

**Institutional Review Board Statement:** Not applicable.

**Informed Consent Statement:** Not applicable.

**Data Availability Statement:** Not applicable.

**Conflicts of Interest:** The author declares no conflict of interest.

## Abbreviations

| Notations | Description of notations |
|---|---|
| M | Number of virtual machines (VM) |
| N | Number of tasks arrived at given instance time |
| $VM_l$ | VM with lowest completion time |
| $VM_h$ | VM with highest completion time |
| $dct_{max}$ | Maximum completion time difference |
| $X^t_k$ | Current particle position for particle k at iteration t |
| $pBest^t_k$ | Best distribution of tasks into heterogeneous VMs for particle k at iteration t |
| $Optsol$ | Optimal solution |
| $gBest^t_k$ | global best distribution of tasks into heterogeneous VMs for particle k at iteration t |
| $F(gBest^t_k)$ | Fitness value of gBest |

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
