# Peer review of "Optimization of Load Balancing and Task Scheduling in Cloud Computing Environments Using Artificial Neural Networks-Based Binary Particle Swarm Optimization (BPSO)"

_sustainability, doi:10.3390/su141911982_

Round 1

Reviewer 1 Report

Here are my thoughts:

1- The paper's weak organization makes it difficult to follow.

2-Background and gap are missing from the abstract section.

3- The English language is of inferior quality.

4-The introduction hierarchy is missing. The context, gap, how, why, and outcomes were all overlooked.

5-The contribution of the study should be clearly stated in the abstract part and in the introduction section with bullet points.

6-  More references are needed in the introduction section. In the introduction part, include the following crucial works:

https://doi.org/10.1002/dac.4474 , https://link.springer.com/book/10.1007%2F978-3-030-18468-1 , https://doi.org/10.1108/K-12-2020-0909 , https://peerj.com/articles/cs-539/, 

7-If you use equations from other papers, you must properly cite them.

8-There is a need for additional comparisons with current works.

9-Quality of Figs and Tables must be improved.

10-After Section 4, fill out this section with one or two sentences about subsections.

11-The conclusion section is insufficient. It is necessary to mention the paper's limitations and flaws.

Author Response

No.

Comments from Reviewer-1

Reply and modifications as per reviewer’s comments

1.       

The paper's weak organization makes it difficult to follow.

Thankyou for pointing out this concern. We have tried to simplify the organization of the paper and We have also added a paragraph about the organization of our paper in the updated file.

2.       

Background and gap are missing from the abstract section.

Background and gaps are added in the abstract in the updated file.

3.       

The English language is of inferior quality.

The revised manuscript has been thoroughly revised for grammatical corrections and English mistakes

4.       

The introduction hierarchy is missing. The context, gap, how, why, and outcomes were all overlooked.

As per your suggestions sir, we have added  all the raised concern in the introduction part .

5.       

The contribution of the study should be clearly stated in the abstract part and in the introduction section with bullet points.

We have added a section 1.1. as contributions and also in abstract we have added lines about contribution.

6.       

More references are needed in the introduction section. In the introduction part, include the following crucial works: https://doi.org/10.1002/dac.4474 , https://link.springer.com/book/10.1007%2F978-3-030-18468-1 , https://doi.org/10.1108/K-12-2020-0909 , https://peerj.com/articles/cs-539/, 

Thank you for suggestion some great research articles we have added

 https://doi.org/10.1002/dac.4474 ,

https://link.springer.com/book/10.1007%2F978-3-030-18468-1 ,

 https://doi.org/10.1108/K-12-2020-0909 ,

 https://peerj.com/articles/cs-539/,

 on [45],[47],[49],[50].

Moreover, we also have added article on

·        Deep Q-Learning

·        Applications of M/L

·        COMPUTING HISTORY

7.       

If you use equations from other papers, you must properly cite them.

Thank you so much fir the suggestion, We have cited the paper from which we have used the equations.

8.       

There is a need for additional comparisons with current works.

Thankyou so much for your suggestion sir, as our focus is on bio-inspired algorithm so we compared with popular algorithm like PSO,IBPSO, Metaheuristic, Heuristic .Adding more algorithm is our future work

9.       

Quality of Figs and Tables must be improved.

We have updated the quality of figure and tables in the update manuscript

10.    

After Section 4, fill out this section with one or two sentences about subsections.

As per your suggestions sir we have added some line after section 4.

11.    

The conclusion section is insufficient. It is necessary to mention the paper's limitations and flaws.

Thanks, you for your suggestions sir we have added limitations and flaw in the conclusion section

Reviewer 2 Report

The paper has several major issues regarding novelties, writings, and contributions. I have the following comments as follows:

1.       There are several important findings in the literature in this direction. Therefore, it is important to obtain the novel findings of this research. There must be a comparative study for transportation purpose with the following articles (Assessing the financial rеsоurсе curse hypothesis in Iran: Thе nоvеl dynаmiс АRDL approach; Application of the artificial neural network with multithreading within an inventory model under uncertainty and inflation; Geometric programming solution of second degree difficulty for carbon ejection controlled reliable smart production system) to show the major contributions and findings.

2.       Keywords should be perfect. The abstract should contain the details of the study and the findings in a very constructive way.

3.       The introduction should be based on the exact research gap, and the literature review should be based on the specific keywords-based review, and finally, make an author's contribution table to show the novelty and effectiveness of the study. Show all referenced papers in the table to show the contribution of this study.

4.       Please write the significant findings in conclusions. Do not mention all assumptions which have been indicated within the model.

5.       What is the data source of the numerical experiment? Please mention that the data is from industry or literature, i.e., accurate data or artificial data.

6.       Conclusions should be updated with more findings, limitations, and future extensions.

Author Response

No.

Comments from Reviewer-2

Reply and modifications as per reviewer’s comments

1.       

There are several important findings in the literature in this direction. Therefore, it is important to obtain the novel findings of this research. There must be a comparative study for transportation purpose with the following articles (Assessing the financial rеsоurсе curse hypothesis in Iran: Thе nоvеl dynаmiс АRDL approach; Application of the artificial neural network with multithreading within an inventory model under uncertainty and inflation; Geometric programming solution of second degree difficulty for carbon ejection controlled reliable smart production system) to show the major contributions and findings.

Thank you for suggestion some great research articles we have added

Thе nоvеl dynаmiс АRDL approach  [54]

Application of the artificial neural network [56]

Geometric programming solution [59]

Moreover, we also have added article on

·        Intelligent (O2O)

·        time-dependent deterioration under fuzzy learning

2.       

Keywords should be perfect. The abstract should contain the details of the study and the findings in a very constructive way.

In the updated manuscript we have updated the abstract as per your suggestions.

3.       

The introduction should be based on the exact research gap, and the literature review should be based on the specific keywords-based review, and finally, make an author's contribution table to show the novelty and effectiveness of the study. Show all referenced papers in the table to show the contribution of this study.

Thank you for your suggestions sir we have added a new section 1.1 as contribution in the introduction section.

4.       

Please write the significant findings in conclusions. Do not mention all assumptions which have been indicated within the model.

As per your suggestion we have updated the conclusions of the manuscript.

5.       

What is the data source of the numerical experiment? Please mention that the data is from industry or literature, i.e., accurate data or artificial data.

In our research we have used the CloudSim-3.0.3 simulator to assess a proposed algorithm's performance. Additional data was created at random by the simulator itself and used in the testing. It was possible to write all of the Java code and execute it concurrently on an Intel Core i5–6500 PC running at 3.2 GHz with 4GB of RAM. Table 4 lays out the components and parameters of our model. Data source detail I mentioned in section 4.1

6.       

 Conclusions should be updated with more findings, limitations, and future extensions

Thanks, you for your suggestions sir we have added limitations and flaw in the conclusion section

Reviewer 3 Report

The author proposed BPSO-based ANN for load balancing and task scheduling optimization in cloud computing environments and demonstrated such combination outperforms other approaches. However, some revision are required before pulication consideration:

(1) The format of this pdf version is very strange, i.e., Text is on the left side while the figures and tables are on the right side. In addition, it is suggested to significantly improve the quality of all figures and tables, which looks blurry or sometimes font sizes are not the same.

(2) Some references, e.g., apply the ABC to enhance the NN's performance which outperforming the PSO algorithm should be referred and compared. In addition, the recent studies also applied the Stochastic Gradient Decent (SGD) with batch sampling to optimize the weights and biases in deep learning work. Or applied the physics-guided constraints to improve the optimzaition performance. Why the author not use and it is suggested to refer and discuss.

Artificial bee colony Based Bayesian Regularization Artificial Neural Network approach to model transient flammable cloud dispersion in congested area.

Probabilistic real-time deep-water natural gas hydrate dispersion modeling by using a novel hybrid deep learning approach

Real-time natural gas release forecasting by using physics-guided deep learning probability model

(3) Could the author give some quantitative results in the conclusion part?

(4) Overall, this is a good work since the author conduct many comparsions with the existed approaches. Hope the above suggestions are useful to improve this work.

Author Response

No.

Comments from Reviewer-3

Reply and modifications as per reviewer’s comments

1

The author proposed BPSO-based ANN for load balancing and task scheduling optimization in cloud computing environments and demonstrated such combination outperforms other approaches. However, some revisions are required before publication consideration:

Dear Reviewer, thank you for your valuable suggestion, we appreciate your support and guidance.

2

The format of this pdf version is very strange, i.e., Text is on the left side while the figures and tables are on the right side. In addition, it is suggested to significantly improve the quality of all figures and tables, which looks blurry or sometimes font sizes are not the same.

In the updated manuscript we have formatted it properly and we have improved the figure and tables.

3

Some references, e.g., apply the ABC to enhance the NN's performance which outperforming the PSO algorithm should be referred and compared. In addition, the recent studies also applied the Stochastic Gradient Decent (SGD) with batch sampling to optimize the weights and biases in deep learning work. Or applied the physics-guided constraints to improve the optimization performance. Why the author not use and it is suggested to refer and discuss.

Artificial bee colony Based Bayesian Regularization Artificial Neural Network approach to model transient flammable cloud dispersion in congested area.

Probabilistic real-time deep-water natural gas hydrate dispersion modeling by using a novel hybrid deep learning approach

Thank you for suggestion some great research articles we have added

1.      ABC to enhance the NN's performance [58]

2.      Stochastic Gradient Decent (SGD) [60]

3.      physics-guided deep learning probability [61]

Moreover, we also have added article on

·        Faster R-CNN technique

·        Methodological improvements

4

5

Could the author give some quantitative results in the conclusion part?

Thank you for your suggestions, we have added some quantitate results in conclusion section.

Overall, this is a good work since the author conduct many comparisons with the existed approaches. Hope the above suggestions are useful to improve this work.

Dear Reviewer, thank you for your valuable suggestion, we appreciate your support and guidance.

Round 2

Reviewer 1 Report

It can be accepted.

Reviewer 2 Report

The paper can be accepted for publication.

Reviewer 3 Report

All the comments have been addressed well. Thanks.